# Remote Monitoring of Canine Patients Treated for Pruritus during the COVID-19 Pandemic in Florida Using a 3-D Accelerometer

**DOI:** 10.3390/ani13243875

**Published:** 2023-12-16

**Authors:** Michael Canfield, Robert P. Lavan, Timberly Canfield, Tonya Springer, Rob Armstrong, Gal Gingold, Jennifer Thomas, Bridgette Sampeck

**Affiliations:** 1Animal Dermatology South, 7741 Congress Street, New Port Richey, FL 34653, USA; drmc@ahrpvet.com (M.C.); twc2332@gmail.com (T.C.); tclark@ahrpvet.com (T.S.); bsampeck7@gmail.com (B.S.); 2Merck & Co. Inc.,126 E. Lincoln Ave, Rahway, NJ 07065, USA; 3Merck Animal Health, 128 E. Lincoln Ave, Rahway, NJ 07065, USA; robert.armstrong@merck.com; 4Sure Petcare, Ground Floor, Building 2030, Cambourne Business Park, Cambourne CB23 6DW, UK; gal.gingold@merck.com

**Keywords:** pruritus, 3-D accelerometer, monitoring, allergic skin disease, canine atopic dermatitis

## Abstract

**Simple Summary:**

This study describes the use of a motion-detecting device (3-D accelerometer) to remotely monitor therapy in dogs who had been diagnosed and treated for itch by a dermatologist during a prolonged period that included the 2020–2023 COVID-19 pandemic. Ensuring dog owner adherence to recommended protocols, obtaining early warning of flare-ups, and maintaining close monitoring of patient condition are factors that can dramatically improve the outcome. Movement monitoring devices linked with data recording apps combined with real-time communication with the veterinary practice can help to address these challenges and lead to improved medical management.

**Abstract:**

The medical management of chronic canine pruritic dermatologic conditions is challenging and often frustrating. This is a report that shows one way of aiding the management of pruritic dogs using a remote monitoring device. It is often difficult for veterinarians to get dog owners to return to the clinic once a dog is treated. It is possible that a 3-D accelerometer device could provide information to the clinic staff on the success or failure of a pruritus treatment plan while the dog was cared for at home. Eighty-seven dogs and their owners came to a Florida dermatology specialty clinic or its general practice hospital to be evaluated and treated for pruritus. An ANIMO^®^ 3-D accelerometer was placed on the collar of dogs diagnosed and treated for pruritus. Dogs that completed this study were monitored for 120 days (4 months). The ANIMO smart phone application monitored a dog’s daily scratching, shaking, sleeping, activity, and resting and summarized this information in a daily report visible on the pet owner’s smart phone. An additional variable (grooming minutes per day) could be seen by the study team that was not yet available in the app. The use of a 3-D accelerometer enabled veterinarians to continuously monitor dogs at home when they were being treated for itching. Clinic staff kept in touch with the owners by phone and could change therapy or bring the dog back for a recheck if problems were seen. Daily reports were combined into line charts that showed plots of scratching, shaking, grooming, and sleeping over four months. Veterinarians were able to remotely monitor dogs that had been treated for pruritus for up to four months through use of a collar-borne monitoring device. Dog owners and clinic staff used the daily summaries accessible through a smart phone application. Dogs seemed to tolerate the device well because of its small size, light weight, long battery life, and unobtrusive nature.

## 1. Introduction

Dog dermatologic diseases are frequently associated with pruritus that leads to skin self-trauma and notable skin injury and is associated with impaired rest profiles for both the owner and the dog, although rest profile changes are not readily measured. The dog’s motor responses to pruritus produce characteristic motions that can be detected with a monitoring device. ANIMO^®^ (Sure Petcare, a division of Merck Animal Health) is a 3-dimensional accelerometer device developed in 2018 to detect changes in animal motion (secondary to particular behaviors) and enable the tracking of these over time. ANIMO clips onto the dog’s collar, has a battery life that is reported to last up to 6 months, and can monitor motions associated with many behaviors, including pruritus, and response to therapy for pruritus over an extended period.

The ANIMO device was developed following a long period of validation studies in which dogs were continuously monitored with video while wearing a 3-D accelerometer on the collar. These observations, coupled with observations from dog owners, allowed the development of proprietary algorithms to increase the reliability of detecting specific behaviors.

Over the last 10 years, several accelerometer devices were introduced to digitally capture the motion of individuals in several animal species. A 3-D accelerometer ear tag for cattle (Allflex, Merck Animal Health, Rahway, New Jersey) was recently introduced, and this device can detect rumination [1]. Other studies report the use of 3-D accelerometers, including in dogs to detect a variety of health-related movements (arthritis [2] and seizures [3]); in cats (sleep quality [4] and physical activity [5]); in cattle (Bovine Viral Diarrhea [6], bull breeding [7], signal pre-processing [8], lying behavior [9], behavior on pasture [10], and grazing [11]); in chickens (motion intensity [12] and caged bird behavior [13]); in horses (motor behavior [14]); in sheep (grazing and rumination [15]); in arctic muskox (breeding and life events [16]); and in dolphins (startle responses [17]).

ANIMO measures canine movement in 3-dimensions and, when placed on a dog’s collar, produces a constant stream of data-specifying motions made by the dog. These can then be analyzed with data management algorithms and classified into different types of activities, including standard daily activities such as sleep quality, active time, resting time, and calories burned and potentially adverse or undesirable behaviors including barking, scratching, and shaking. When undesired behaviors are recorded over time periods beyond established norms, an alert is sent to the smart phone data presentation application (“app”) after the software is installed on a handheld smart phone. In addition, these alerts can be further shared with others, including the study center and investigator(s), through the smart phone application.

The primary objectives of this study were to evaluate the potential medical value of ANIMO-measured motions in dogs under treatment for pruritus for management of these cases, including the potential for early warnings of recurrence (‘flare up’) in dogs with pruritus. Flare up is a common phenomenon in dogs with chronic pruritic skin disease, and early detection can improve the prognosis for case management, including early intervention to reduce self-trauma.

## 2. Materials and Methods

The clinical veterinarians in this study examined, tested, and diagnosed dogs presenting with pruritus and then recommended treatment according to their best clinical judgement within routine daily function of a referral veterinary dermatology hospital. Written informed consent was obtained from all participants before enrollment in the trial, and this study was initiated by attaching the ANIMO to the dog’s collar. Owner confidentiality was strictly maintained. The ANIMO device is a small (1.45 inch (3.68 cm) diameter × 0.45 inch (1.14 cm) depth; 0.78 oz (22.11 g)) round waterproof device (reported IP67 protected from immersion in water with a depth of up to 3.3 feet (1.0 m) for up to 30 min) that is mounted on the dog’s collar and functions with a CR 2032 battery that reportedly lasts up to 6 months. Acceptable collar widths range from 0.5 inches (1.27 cm) up to 1.25 inches (3.18 cm). It records specific acceleration in three dimensions when fixed to a dog’s collar. The manufacturer indicates that the ANIMO will measure the daily amount of scratching, shaking, sleeping, daily activity, and resting time by a dog and is also evaluating a new experimental variable (time spent grooming) which was not included in the pet owner’s report but was available to the researchers. ANIMO uses a combination of Bluetooth and unique algorithms to monitor the dog’s movement. When first attached to the dog, the ANIMO interpretive algorithm takes up to 2 weeks to establish baseline behavior for the dog (Sure Petcare (SPC); https://www.surepetcare.com/en-us/animo/animo-classic, accessed on 21 February 2023).

This study was conducted in Florida, a high-risk region for pruritic disease in dogs, including flea infestation and allergic skin disease. This was a non-interventional prospective cohort study of dogs presented to a specialty dermatology animal hospital for pruritus of any cause. All dogs were seen by a veterinarian and had their source of pruritus diagnosed and treated. Dogs were excluded from this study if the owner did not have access to a smart phone, if they could not use the ANIMO app, or if they were seen at the dermatology clinic for a non-pruritic condition. Also, a dog could be discontinued from this study when there was evidence that the collar was not monitoring the dog for an extended period (over 5 days) or where more than 5 consecutive days of data were lost. There were no restrictions on dog age, weight, breed, gender, or neutering status.

The demographic data were collected as dogs were enrolled and were placed in an Excel^®^ (Microsoft, Redmond, WA, USA) spreadsheet that was later summarized by the veterinary clinic staff at the end of this study. ANIMO data were monitored daily, and a weekly report was produced to provide a method to remind delinquent pet owners to sync their smart phone with ANIMO as well as to show the clinic staff which dogs were receiving shaking or scratching alarms. The clinic staff contacted the referring veterinary clinics to find out missing information such as past and current flea and tick medication or prior diagnoses.

ANIMO was attached to the dog’s neck collar and was remotely monitored at home through the ANIMO app (Sure Petcare—ANIMO, Activity and Behavior Monitor) installed on the owner’s smart phone. After fitting ANIMO at the veterinary hospital, dogs were followed for an initial 14 days of calibration to allow ANIMO to normalize to the dog’s individual movement profile. All enrolled dogs’ behaviors were tracked over the subsequent 4-month period.

The owner paired their smart phone with the app on the dog’s collar (also known as syncing) every few days to upload stored ANIMO data. Up to 2 weeks of continuous data could be stored in the ANIMO device between syncing. Once uploaded, all data were stored on remote internet servers (also known as in the cloud) for monitoring and were available to the owner and hospital staff through the smart phone app.

This study evaluated the potential for ANIMO to provide a health profile to veterinarians that could be used to monitor response to treatment, document therapeutic progress, and capture evidence for relapse. Additionally, dog owners could see the change in their dogs’ movement profiles following treatment for dermatologic disease. The variables reported by ANIMO are listed in Table 1.

The research team also provided a summary of the previous weeks’ monitoring data to the veterinary clinic staff for each dog. This summary included a list of scratch and shake alerts, notification when an owner was delayed in syncing their dog’s data, updates on devices needing a battery change, and reports of dogs not wearing the ANIMO device. Clinic staff used this weekly report to review the status of enrolled dogs and determine whether a call to the owner was needed for an update or action. When a scratch or shake alert was recorded, owners were asked if they were seeing increased scratching or shaking and then had an opportunity to set up a recheck visit, wait and watch, and/or adjust therapy.

Veterinary hospital staff regularly examined daily trend reports for monitored dogs and then reached out to owners of dogs with results that suggested that pruritus was more severe or more frequent. Contacted owners could elect to continue monitoring at home; to work with the veterinarian to institute a change to therapy; or to return to the practice for a recheck visit. Contacting the pet owner soon after the alert (within a day or two) supported the health status of the dog while the dog’s behavior was still fresh in the owner’s memory. Since this study was completed, the ANIMO software (version 2.7.2.0) was upgraded to offer a daily and weekly summary of the data.

At the conclusion of this study, each dog owner completed a short survey on their ANIMO experience. Additionally, selected daily variables were compiled by the authors into a line chart for each dog to visualize the amount of scratching, shaking, grooming, and sleeping over 4 months of monitoring. The line chart was not available as part of the ANIMO app at the time of this study but is being considered for possible development.

Line charts were developed to visually represent 4 months of information in a single view. This information was compiled from the ANIMO app by examining data fields on consecutive days. Specific calendar dates were aligned with study days (day 1 to day 120) to allow association between events in the medical notes (communication with the pet owner or initiation of a treatment change, for example) and activities in the 4-month line chart. The wave amplitude pattern (represented by minutes per day per variable) for scratching, shaking, and grooming behavior and the number of shake or scratch alerts was examined for each dog and remained consistent for individuals during the 4-month monitoring window.

National COVID-19 restrictions were placed by the CDC and U.S. Federal Government in March 2020. As a result, the veterinary clinics in this study met patients and owners curbside for more than a year to reduce possible exposure to the COVID-19 virus. Restrictions required face mask use and appropriate spacing of all clinic visitors. COVID-19 restrictions, on occasion, reduced owner willingness to return to the clinic for rechecks, and this led to a prolonged close-out period. There was no COVID-19 impact on the four-month ANIMO monitoring period, once started, because this could be completed remotely without direct contact with either the dog or owner and the clinic staff.

Daily mean values for variables captured by ANIMO were compared using a two-sample *t*-test assuming unequal variances.

## 3. Results

Eighty-seven dogs were diagnosed with pruritic skin disease and subsequently completed four months of monitoring by ANIMO. This study was initiated on 1 April 2020 with a rolling enrollment period that lasted for 16 months, with the last dog completed on 16 July 2021.

A summary of the mean dog age and body weight by pruritus diagnosis is presented in Table 2. Dogs were assigned to treatment groups based on the pruritus diagnosis. A convenience sampling method was used, and dogs were accepted into this study as they presented at the clinic for treatment. Most dogs were diagnosed with allergic dermatitis (48%) or atopic dermatitis (38%) and a few were diagnosed with otitis externa (9%) or suspected allergy (5%). The most common breeds brought to the veterinarian for pruritus were Terriers (*n* = 12, 14%), Retrievers (*n* = 11, 13%), Spaniels (*n* = 7, 8%), Pit Bull-type (*n* = 7, 8%), Shepherds (*n* = 6, 7%), Mixed breed (*n* = 5, 6%), Poodles (*n* = 4, 5%), and Bulldogs (*n* = 4, 5%). Genders were equally distributed (male = 44, female = 43), and most dogs were neutered (83/87, 95%).

Owners were able to view a summary of the previous day’s data on their smart phones each morning. While most of the data could be viewed in the smart phone app in real time, scratch and shake alerts could only be seen in the morning summary report for the previous day. Scratch or shake alerts were generated by the device when the behavior occurred more severely (seemed to cluster). There were 537 scratch alerts generated by all dogs over 120 days. Individual dogs in this study generated from 0 to 29 scratch alerts over the study period, with the average dog generating 7.4 scratch alerts over the four-month monitoring period. About 25% of dogs generated no scratch alerts at all, and 75% of dogs generated one or more alerts. When the number of scratch alerts was placed into blocked ranges, it was clear that most dogs diagnosed with allergic dermatitis generated very few (0 or 1–5) scratch alerts over the entire study period (Figure 1). Dogs diagnosed with atopic dermatitis were more likely to generate 1–15 scratch alerts over the same period. The difference in the number of scratch alerts generated between allergic dermatitis and atopic dermatitis was statistically significant (two-sample t-test assuming unequal variances, *t*_(7761)_ = −4.31, *p* < 0.05).

Across all dogs, there were 1209 total alerts (shaking plus scratching) generated in this study over 120 days. Individual dogs in this study generated from 0 to 57 total alerts, with the average dog generating 16.8 total alerts over the 4-month period. Five percent of dogs (*n* = 4) generated no alerts at all and came from the allergic dermatitis group (*n* = 2), atopic dermatitis group (*n* = 1), and otitis externa group (*n* = 1). Ninety-five percent of dogs generated one or more alerts over the study period. The largest proportion of allergic dermatitis dogs generated 1–10 total alerts, with the largest proportion of atopic dermatitis dogs generating 11–20 total alerts. The number of total alerts was not significantly different between dogs diagnosed with atopic dermatitis and dogs diagnosed with allergic dermatitis (two-sample *t*-test assuming unequal variances, *t*_(16677)_ = −1.73, *p* = 0.08; Table 3).

Table 3 is a presentation of the mean (±standard deviation) of each variable captured by ANIMO for all dogs broken out by pruritus diagnosis group. These data demonstrate that the average dog in this study had approximately 6 min of scratching per day, 4 min of shaking, 29 min of grooming, 7 h of night rest, and 80 min of total activity per day. Statistical comparisons were made between the means for allergic dermatitis and atopic dermatitis dogs using a two-sample t-test. Statistical comparisons did not include dogs diagnosed with otitis externa or suspected allergy because of the low number of dogs in these groups. Allergic dermatitis and atopic dermatitis were significantly different for minutes of shaking, minutes of total activity, and minutes of low activity.

At the end of this study, data from daily ANIMO reports were compiled into 120-day line charts which presented the combined data for scratching, shaking, grooming, night rest, and sleep ratio (Figure 2, Figure 3 and Figure 4). Each individual dog had a wave pattern determined by the number of minutes per day per variable. Some dogs made only small amplitude waves (Figure 2), which indicated fewer minutes of daily scratching, for example. Other dogs made moderate amplitude waves (Figure 3) and others made larger amplitude waves (Figure 4), indicating more scratching, shaking, and/or grooming behavior. The wave size pattern is most apparent in the line for grooming behavior but can be easily seen in the scratching and sleep ratio lines as well (Figure 4). The scratch and shake alerts (vertical green and purple lines) indicate periods of more severe (cluster) scratching and/or shaking behavior. While all dogs were being treated for itch, some dogs generated no scratch or shake alerts for the entire monitoring period, suggesting they were more comfortable with their pruritus. The line chart for an individual dog simplifies the observation of changes in daily sleep quality, along with the presence and timing of alerts related to specific, potentially injurious, behaviors including scratching and shaking.

The exit survey was completed by 77 dog owners, although a few dog owners did not answer all questions. Almost all (96% or 74/77) responders felt that they had a good experience with ANIMO. Most owners reported checking the app on a daily basis (62%, 47/76), with fewer owners checking on a weekly basis (36%, 27/76) or monthly basis (2%, 2/76). More than 80% of responders named the activity information (low/medium/high, graphs, and goal setting), sleep quality, and daily resting time as well as minutes of scratching and shaking as the most useful information provided by the app. Most owners indicated that they did receive warnings for scratching or shaking (71%), although fewer noted a decreased sleeping score (33%) at some point during this study. A high proportion (90% 69/77) of participants would recommend ANIMO to a friend. Several owners indicated that they felt encouraged to increase their dog’s exercise, as well as their own exercise, in order to meet a goal based on the reported amount of daily activity.

## 4. Discussion

Accelerometer movement monitoring by a collar-attached device provided valuable medical management data for dogs presenting with pruritus to a veterinary dermatological referral practice. Owners and veterinarians received information that assisted in understanding the movement pattern for these dogs over a 4-month monitoring period. In particular, the potential to provide an early alert of pruritus “flare up” offered notable value and provided an opportunity for increased communication between the dog owner and clinic staff.

This study was conducted against the background of a global COVID-19 pandemic that began in 2020 and continued into 2023, at the time of this writing. This timing was unexpected and coincidental; however, it provided a unique opportunity to evaluate a remote movement monitoring device to follow therapy for dogs being treated at home for pruritic skin disease. The battery life of the device was observed to provide up to 6 months of continuous monitoring, which allowed most dogs to complete this study without a battery change.

To deliver uninterrupted data collection, the owner needed to ensure that the phone application automatically or manually synchronized with the ANIMO software more frequently than every 14 days. Syncing uploads data stored in the collar-mounted device to the cell phone for processing followed by the generation of notifications to the owner, the clinic staff, and the veterinarian. This remote monitoring is not dependent on how close the owner lives to the veterinary clinic because data were uploaded to specific internet servers (“the cloud”) whenever the owner’s cell phone had access to a cellular phone network.

The ANIMO 3-D accelerometer measures movement that is translated into specific movement parameters, and this report focuses on those expected to be more related to itch. Owners were able to see other variables (Table 1) of interest, including reduced nightly sleep quality, daily energy expenditure, and increased grooming, scratching, or shaking minutes. ANIMO detected and recorded these behaviors at frequent intervals, including at night when the owner was usually not observing the dog. Sleep quality assessment for the dog may translate to owner sleep quality since many of the dogs sleep with or near the owners. In one case, the clinic staff called the pet owner to inquire about the dog’s episodes of interrupted sleep. The owner reported that the dog was getting up every time she experienced insomnia, and the device was able to report this.

The veterinarian can also use ANIMO to monitor their patient, in cooperation with the dog owner. The veterinarian should set pet owner expectations because data provided by the ANIMO are not diagnostic but help to monitor behaviors indicating response to treatment and potential recurrence of detectable clinical signs. The owner gains from veterinary assistance to interpret reported results, with the timing of communication dependent on the speed of change and severity of clinical signs. The occurrence of reports alerting the owner and veterinarian about deviations from the normal movement standard for the dog can be a trigger for further communication. As a case example, there was a patient who experienced an otitis flare while the primary caregiver was out of town, and the accelerometer alert triggered the owner to inquire about the patient, resulting in additional treatment being prescribed in advance of the return to her veterinarian.

Veterinary hospital staff already communicate regularly with owners; however, ANIMO reports offer an innovative change because the dog’s behavior is the stimulus for the veterinary hospital staff to contact the owner for an update. This inverts the traditional approach where the owner would contact the veterinary hospital about an observed behavior of concern. The requirements of this study meant that owners were contacted by the clinic staff to investigate a concern or question arising from reported ANIMO data. One veterinary hospital staff member was assigned as the primary contact for ANIMO owners, providing continuity in the communication. This veterinary hospital found using text messaging to owners was more effective than voice calling or emailing. All communication was documented in the medical record at the time of occurrence.

On occasion, owners removed ANIMO from the dog’s collar, usually at the time of bathing, swimming, or visits to the groomer. Sometimes owners needed a reminder call to put the collar back on the dog to ensure no loss of data. ANIMO holds 14 days of data internally before resyncing without data loss. Dog owners were sometimes reminded by the clinic staff to resync the data to the smartphone.

To observe the entire four months of sequential data in this study, daily data summaries for each dog were compiled into a single line chart for the full four-month monitoring period, showing scratching, shaking, grooming, and sleeping trends. These charts were created at the end of this study and are being considered for future software iterations. A variable, “grooming” (in minutes per day), was developed and not yet available to device users but was included in the four-month line chart. Four continuous months of data in one chart makes clear the uniqueness of each dog’s movement patterns and the ANIMO interpretation of this movement. Dogs generated scratch or shake lines with small, medium, or large amplitude waves in a pattern reflecting the minutes of behavior per day for each dog over the course of this study. These multi-month line charts allow the veterinarian to evaluate changes in the baseline for each variable that identify potential episodes of increasing pruritus or show signs of treatment response and increased comfort. ANIMO monitoring demonstrated a significant monitoring value for these patients in this study through the early identification of flares in itching.

The dogs in this study would not be considered normal because they were all being treated for itch. We are not aware of any prior published data on how ANIMO reports these variables for dogs which are non-pruritic and essentially normal. It was interesting that dogs with allergic dermatitis on average experienced about 1 min more of shaking per day (about 20%) and about 10 min less total activity per day (also about 20%) compared to dogs diagnosed with atopic dermatitis. There was no difference seen between these two groups in average scratching minutes per day.

Scratch or shake alerts seem to be a warning about more intense scratching or shaking behavior and were used by the clinic staff as one reason to contact the dog owner about a need to increase monitoring and possibly adjust therapy. Dogs diagnosed with atopic dermatitis had significantly more scratch alerts when compared with dogs diagnosed with allergic dermatitis, which may be an indicator that atopic dogs need to be monitored more closely and possibly treated longer and more often than allergic dermatitis dogs.

Daily ANIMO activity reporting motivated some owners to increase their daily activity with their dog. Several owners mentioned that if the dog did not reach an identified activity goal, then they would go out at night to walk with the dog to meet or exceed the calorie goal. This was an unexpected positive outcome for owners participating in this study and could provide a positive benefit in their own physical and mental health. Some owners with multiple conflicting pressures may not be as observant of their dog, and ANIMO provided evidence to owners regarding their dog’s ongoing itch behavior. Several observant owners noted changes on their daily ANIMO activity report and were able to work with the veterinarian to receive early intervention for their dog’s skin disease in the initial stages of a flare up. An example is included to show how information from the app flagged an increase in itching. A male neutered 9-year-old 6 kg pug cross dog was previously diagnosed with atopy that was being treated at home. ANIMO provided warning of a flare up in his pruritic condition before this was observed by the owner. Communication between the veterinarian and owner was initiated, leading to the modification of the pruritus management protocol and improvement in clinical signs. The owner did not have to come back to the clinic to obtain additional treatment for the dog.

## 5. Conclusions

Accelerometer use provides an additional layer of monitoring for pruritic skin disease, especially when close monitoring is difficult or neglected. The use of a remote monitoring device allows veterinary hospital staff to see how dog owners are progressing with managing their dog’s pruritus at home. The ability of the dog owner, clinic staff, and/or veterinarian to observe the amount of daily activity, sleeping, scratching, and shaking (and grooming when available) provides a more complete picture of the dog’s response to therapy. Recording exercise also drives many dog owners to increase their own exercise when they increase the exercise of their pets to meet daily goals. The use of movement monitoring devices is a potentially valuable new aid for the management of chronic pruritic skin disease of dogs because flare ups of itch with scratching are common sequelae of skin disease.

## Figures and Tables

**Figure 1 animals-13-03875-f001:**
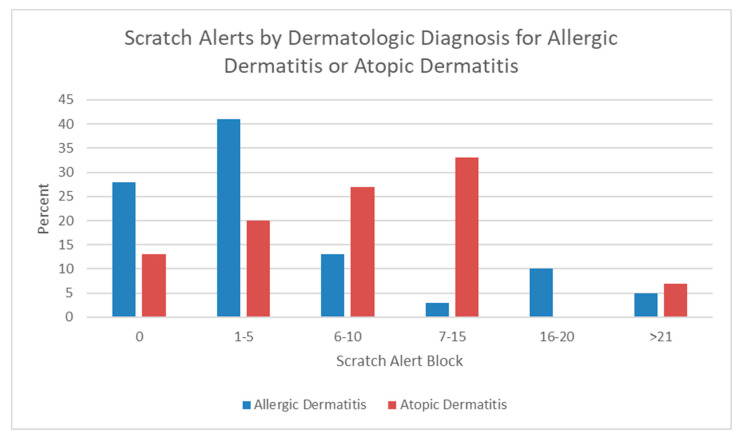
Scratch alerts in blocks by dermatologic diagnosis.

**Figure 2 animals-13-03875-f002:**
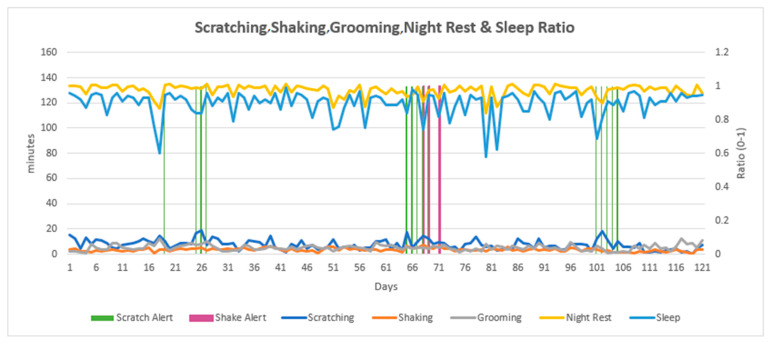
An example of a 4-month line chart for a single dog exhibiting small amplitude waves also showing scratch and shake alerts. The left vertical axis is minutes per day for scratching, shaking, and grooming, and minutes of night rest per day are divided by 3 to fit a similar range seen for other daily measures. The right vertical axis is the sleep ratio per night (light blue line), a units-free assessment with a result between 0 and 1 that assesses the time proportion spent in sleep behavior during the night.

**Figure 3 animals-13-03875-f003:**
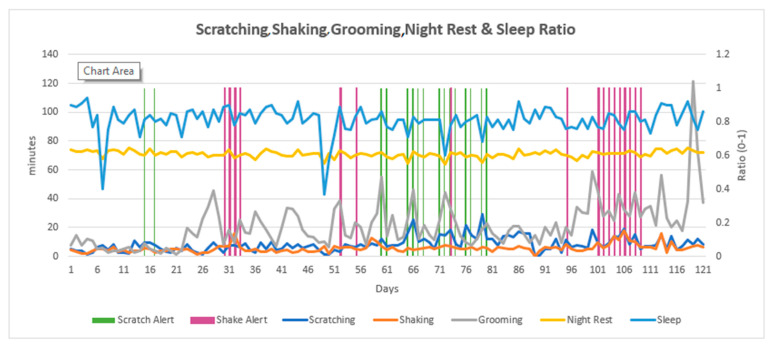
An example of a 4-month line chart for a single dog exhibiting moderate amplitude waves also showing scratch and shake alerts. The left vertical axis is minutes per day for scratching, shaking, and grooming, and minutes of night rest per day are divided by 3 to fit a similar range seen for other daily measures. The right vertical axis is the sleep ratio per night (light blue line), a units-free assessment with a result between 0 and 1 that assesses the time proportion spent in sleep behavior during the night.

**Figure 4 animals-13-03875-f004:**
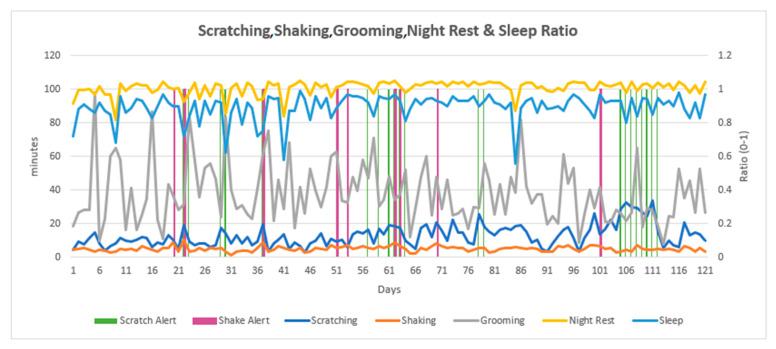
An example of a 4-month line chart for a single dog exhibiting large amplitude waves also showing scratch and shake alerts. The left vertical axis is minutes per day for scratching, shaking, and grooming, and minutes of night rest per day are divided by 3 to fit a similar range seen for other daily measures. The right vertical axis is the sleep ratio per night (light blue line), a units-free assessment with a result between 0 and 1 that assesses the time proportion spent in sleep behavior during the night.

**Table 1 animals-13-03875-t001:** Daily variables provided by ANIMO.

Animo Smart Phone Variable	Units of Measure
Last data sync with the app	Number of days since last sync
Daily activity intensity	Time of day X active/inactive chart
Type of activity (low, medium, or high intensity)	Minutes per day
	Activity broken out per day over weeks/months
Sleep quality (last night, average score)	Percent of sleep time with good sleep
	Minutes awake, restless, normal, deep sleep
Daily resting (night rest, day rest)	Minutes per day or night
	Amount of day rest/night rest for day/week/month/year (minutes)
Wellbeing (scratching, grooming ^1^, shaking per day)	Minutes per day for each variable

^1^ At the time of this study, time spent grooming was an experimental variable that was being tested for potential inclusion in the ANIMO app.

**Table 2 animals-13-03875-t002:** Pruritus diagnosis by dog age and weight.

Pruritus Diagnosis	N	Mean Age ± SD (yrs)	Age Range (yrs)	Mean Weight ± SD (kg)	Weight Range (kg)
Allergic dermatitis	42	6.1 ± 3.1	0.4–12.0	24.6 ± 14.4	2.4–54.6
Atopic dermatitis	33	5.4 ± 2.7	1.8–11.0	23.3 ± 13.9	2.4–53.3
Otitis externa	8	7.5 ± 4.6	1.8–14.0	19.0 ± 12.6	7.1–33.8
Allergy suspected	4	5.6 ± 3.7	1.6–11.0	17.9 ± 11.3	4.0–29.9

**Table 3 animals-13-03875-t003:** Daily mean values (±standard deviation) in minutes per day for variables captured by ANIMO by pruritus diagnosis across 120 days of monitoring.

	Allergic Dermatitis (*n* = 42)	Atopic Dermatitis (*n* = 33)	Otitis Externa (*n* = 8)	Allergy Suspected (*n* = 4)	Overall Mean (*n* = 87)
Scratching	5.8 ± 3.5 ^a^	6.8 ± 4.2 ^a^	5.4 ± 3.4	8.9 ± 4.9	6.3 ± 3.9
Shaking	4.1 ± 1.6 ^a^	3.4 ± 1.0 ^b^	4.2 ± 1.9	3.5 ± 1.3	3.8 ± 1.4
Grooming	29.4 ± 14.9 ^a^	31.6 ± 18.7 ^a^	21.8 ± 9.8	27.7 ± 7.4	29.4 ± 15.9
Night rest	402.3 ± 60.6 ^a^	424.9 ± 58.5 ^a^	433.6 ± 65.7	480.9 ± 41.1	417.4 ± 61.5
Sleep ratio	0.84 ± 0.07 ^a^	0.85 ± 0.07 ^a^	0.85 ± 0.11	0.87 ± 0.05	0.85 ± 0.07
Total activity	75.5 ± 28.5 ^a^	88.9 ± 24.9 ^b^	69.6 ± 24.5	91.4 ± 15.3	80.8 ± 27.0
Low activity	48.2 ± 17.7 ^a^	57.2 ± 19.8 ^b^	46.6 ± 17.9	50.3 ± 13.0	51.5 ± 18.7
Medium activity	3.7 ± 6.0 ^a^	3.4 ± 2.8 ^a^	2.7 ± 2.5	6.3 ± 8.6	3.6 ± 4.8
High activity	23.6 ± 13.8 ^a^	28.3 ± 14.3 ^a^	20.3 ± 8.0	34.8 ± 15.4	25.6 ± 13.8

The means for allergic dermatitis and atopic dermatitis were compared statistically using a two-sample *t*-test. Values with different superscripts differ significantly by *p* < 0.05.

## Data Availability

The data compiled for this study are not openly available as they are part of the patient’s medical record.

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
