# Peer review of "Remote Monitoring of Canine Patients Treated for Pruritus during the COVID-19 Pandemic in Florida Using a 3-D Accelerometer"

_animals, 2023, doi:10.3390/ani13243875_

Round 1

Reviewer 1 Report (Previous Reviewer 1)

Comments and Suggestions for Authors

I found the paper to be greatly improved.  I have no further comments.

Author Response

Thanks to our Reviewer for all the time and care they put into the document. Your input made it better. 

Reviewer 2 Report (Previous Reviewer 2)

Comments and Suggestions for Authors

Statistical analysis, even if it is as simple as descriptive or t-tests, should be described under Materials and Methods in a separate section.

Table 3- what are the units of each behavior? Mean values of what? Minutes? Seconds? Each table should stand alone.

I cant see the line numbering anymore.. please avoid terms in quotation, like: The dogs in this study would not be considered “normal”.

In “wave size pattern is most apparent in the line for grooming behavior but can be easily seen in the scratching and sleep ratio lines as well.” – you have to cite (Figure 4) at the end.

Comments on the Quality of English Language

Minor grammatical mistakes. Please avoid terms in quotation. 

Author Response

Response to Reviewer 2 for manuscript entitled “Remote monitoring of canine patients treated for pruritus during the COVID pandemic in Florida using a 3-D accelerometer”.

November 25, 2023

Authors Response:         We thank the reviewer for their patience and diligence in providing excellent guidance for this manuscript.

Statistical analysis, even if it is as simple as descriptive or t-tests, should be described under Materials and Methods in a separate section.

Authors Response: The last paragraph in Materials and Methods has a statement about the use of two-sample t-tests assuming unequal variances to compare variables.

Table 3- what are the units of each behavior? Mean values of what? Minutes? Seconds? Each table should stand alone.

Authors Response: Table 3 units for every cell is minutes per day. This was added to the Table 3 title.

I cant see the line numbering anymore.. please avoid terms in quotation, like: The dogs in this study would not be considered “normal”.

Authors Response: Continuous line numbering was added which now appears on the right margin. Quotation marks were removed from the word normal on page 12.

Reviewer 3 Report (Previous Reviewer 3)

Comments and Suggestions for Authors

I would like to thank the authors for revising their manuscript and to have responded to comments.

All comments have been properly and satisfactorily addressed.

The last comments I  have for the revised versions are:

Table 2. Usually weights are reported in kg (not in lbs).

Table 3. Please add the units of the ‘values’ (minutes? hours?...).

I have assumed that the second paragraph starting by ‘ 3. Results’  will be removed from the final version (redundant).

Author Response

Response to Reviewer 3 for manuscript entitled “Remote monitoring of canine patients treated for pruritus during the COVID pandemic in Florida using a 3-D accelerometer”.

November 25, 2023

Authors Response:         We thank the reviewer for their patience and diligence in providing excellent guidance for this manuscript.

I would like to thank the authors for revising their manuscript and to have responded to comments.

All comments have been properly and satisfactorily addressed.

The last comments I  have for the revised versions are:

Table 2. Usually weights are reported in kg (not in lbs).

Authors Response: We converted lbs to kgs in Table 2.  

Table 3. Please add the units of the ‘values’ (minutes? hours?...).

Authors Response: Table 3 units for every cell is minutes per day. This was added to the Table 3 title.

I have assumed that the second paragraph starting by ‘ 3. Results’  will be removed from the final version (redundant).

 Authors Response: The second paragraph of the Results discusses the demographics of the enrolled patient population. This is the only paragraph where that is mentioned. We assume you are referring to the first results paragraph, which discusses the prolongation of the study by the COVID outbreak. This had already been mentioned at the end of the Materials and Methods so we shortened this section of the Results.

This manuscript is a resubmission of an earlier submission. The following is a list of the peer review reports and author responses from that submission.

Round 1

Reviewer 1 Report

Comments and Suggestions for Authors

I posted a few minor editorial comments in the attached pdf.  My only issue was with the design of the study.  As I understand patients were assigned at the initial clinic visit.  The monitors then collected data over 2 weeks to establish baseline.  My concern is this baseline period was also the recovery period.  Is it a true normal baseline.  My second question from experience with these types of things was no mention of false reports, warnings from the devices?  

Author Response

Reviewer 1:

  1. I posted a few minor editorial comments in the attached pdf. 

Response from authors: Each editorial comment was addressed.

  1. My only issue was with the design of the study.  As I understand patients were assigned at the initial clinic visit.  The monitors then collected data over 2 weeks to establish baseline.  My concern is this baseline period was also the recovery period.  Is it a true normal baseline. 

Response from authors: You are correct that the 2-week period after the initial clinic visit and diagnosis was a period of recovery following treatment. It was necessary to be able to start the study with the animal as good as it could get. The monitoring period of 4 months allowed us to see what happened to each case after the initial therapy. The monitoring baseline needed to be established in order to better understand future excursions from baseline. These dogs were not “normal” and the 2 week monitoring period gave us a chance to see how well their itch was controlled.

My second question from experience with these types of things was no mention of false reports, warnings from the devices?  

Response from authors: Each animal only had one device (ANIMO device) on their collar. In the Materials and Methods, we presented the inclusion and exclusion criteria for dogs and pet owners. Additionally, we discussed reports we received when the ANIMO was removed from the collar (no data was being gathered), when the owner had failed to sync with the smart phone software (data was not being uploaded to the cloud), when the battery life was short or when worrisome data was generated (changes in behavior). All of the warnings were considered on a daily basis for each dog during the entire study period.

Reviewer 2 Report

Comments and Suggestions for Authors

 Remote Monitoring of Canine Patients Treated for Pruritus 2 during the COVID Pandemic in Florida Using a 3-D Accelerometer

Although the idea of the study is interesting and very beneficial to veterinary medicine and pet owners, the study is incomplete. There was no data presented or statistics performed, except for one example table of a patient. To be considered for publication, this work must have some sort of summary table with data of all patients, analyzed with statistics. You cannot draw any conclusions without that. There are some suggestions below.

M&M: what’s the IACUC approval number?

L91: replace “comes” with “functions”

L94: remove “and comes with a 1-year warranty” it doesn’t matter for the study if it has warranty or not, you should not be trying to sell the product.

L99-103: who reported ANIMO’s operational system, the company? Maybe just write “Animo uses a combination of…” if you say that it is reported to use, we expect some sort of reference at the end. Also, you need to reformulate the phrase “ANIMO takes 1-2 weeks to learn..” to more scientific terms. No need to report the retail price.

L108: “diagnosed–as far as possible-and treated” Please use scientific terms. And don’t use hyphens, use commas instead.

Table 1 is very confusing. Is column 1 supposed to be connected with column 2? I don’t understand the titles of each column.

Where is the data and statistics of this work? How can you draw any conclusions that the device worked if you don’t have this essential portion? I see a Table with an example, but there is no Table with a summary of all dogs in the study. What you could do is summarize time spent doing each behavior at baseline, as well as time spent doing each behavior after the treatment was over, to find for example reduction in pruritus that relates with rate of treatment success. Also, correlate clinical outcome with wearables data. These are two essential parts that I do not see in the paper.  

I suggest the authors consult a statistician.

Comments on the Quality of English Language

There is use or jargon and lack of scientific writing. 

Author Response

Response to Reviewer 2

Addressed by Robert Lavan, June 25,2023

Intended for ANIMALS

Title “Remote Monitoring of Canine Patients Treated for Pruritus during the COVID Pandemic in Florida Using a 3-D Accelerometer

Michael Canfield 1, Robert P. Lavan 2,*, Timberly Canfield 3, Gal Gingold 4, Jennifer Thomas 5,
Bridgette Sampeck 6, Tonya Springer 7 and Rob Armstrong 8

  1. Although the idea of the study is interesting and very beneficial to veterinary medicine and pet owners, the study is incomplete. There was no data presented or statistics performed, except for one example table of a patient. To be considered for publication, this work must have some sort of summary table with data of all patients, analyzed with statistics. You cannot draw any conclusions without that. There are some suggestions below.

Response from Authors:See the changes we made. This is a non-interventional observational study and not a comparison of treatment groups. This use of real-world evidence is typical of studies in Outcomes Research. As far as we are aware, this is the first demonstration using a 3-D accelerometer to monitor dogs at home after being treated by a dermatologist. The results demonstrate the potential of this new technology.

  1. M&M: what’s the IACUC approval number?

Response from Authors: IACUC approval is not needed in non-laboratory (field) studies where the natural behavior of animals is being unobtrusively observed. This was a non-interventional cohort study of animals coming to veterinarians for care.

“Studies that involve unobtrusive observation of animals in their natural habitats do not require IACUC oversight. If the study has a potential to cause harm or materially alter the behavior of the animals, then IACUC oversight is required.”

https://research.uci.edu/animal-care-and-use/do-you-need-iacuc-review/#:~:text=Studies%20that%20involve%20unobtrusive%20observation,then%20IACUC%20oversight%20is%20required.

Do You Need IACUC Review? - UCI Office of Research

  1. L91: replace “comes” with “functions”

Response from Authors: Made the suggested change.

  1. L94: remove “and comes with a 1-year warranty” it doesn’t matter for the study if it has warranty or not, you should not be trying to sell the product.

Response from Authors: The warranty statement was removed. The device has been commercially available for a few years. The warranty, battery life and cost would all be things that a practitioner might like to know if they were considering the practicality of adding remote monitoring to their treatment regimen.

  1. L99-103: who reported ANIMO’s operational system, the company? Maybe just write “Animo uses a combination of…” if you say that it is reported to use, we expect some sort of reference at the end. Also, you need to reformulate the phrase “ANIMO takes 1-2 weeks to learn..” to more scientific terms. No need to report the retail price.

Response from Authors: We removed the price reference and altered the sentence about device learning. There is a URL reference at the end of that paragraph for the manufacturer.

  1. L108: “diagnosed–as far as possible-and treated” Please use scientific terms. And don’t use hyphens, use commas instead.

Response from Authors: Corrected this sentence.

  1. Table 1 is very confusing. Is column 1 supposed to be connected with column 2? I don’t understand the titles of each column.

Response from Authors: Table 1 was formatted by the publisher incorrectly. We hope it is clearer now.

  1. Where is the data and statistics of this work? How can you draw any conclusions that the device worked if you don’t have this essential portion? I see a Table with an example, but there is no Table with a summary of all dogs in the study. What you could do is summarize time spent doing each behavior at baseline, as well as time spent doing each behavior after the treatment was over, to find for example reduction in pruritus that relates with rate of treatment success. Also, correlate clinical outcome with wearables data. These are two essential parts that I do not see in the paper.

Response from Authors: We added a data table on dog diagnoses, weight and age with a paragraph with a demographic summary. We do not indicate that the ANIMO readings are sufficient to measure treatment success. Pruritus is multifactorial, and one dog may have multiple allergies. Dogs respond differently to the same medication (ex> cytopoint efficacy over time) and typically require individual dosing. We do not hypothesize that the 3-D accelerometer can measure pruritus reduction; rather, the movement measurement provides an indication that potential pruritus recurrence needs to be addressed through a visit to the veterinary hospital.

  1. I suggest the authors consult a statistician.

Response from Authors:  We did have a staff statistician perform selected statistics, although we did not initially include any of these. We have added the t test analysis for the difference between scratch alerts.

  1. Comments on the Quality of English Language: There is use or jargon and lack of scientific writing. 

Response from Authors: We assume you are referring to the use of computer jargon, like “app” and “cloud”. We have defined the first use of these terms in the text.

Reviewer 3 Report

Comments and Suggestions for Authors

This is a review of the manuscript entitled Remote Monitoring of Canine Patients Treated for Pruritus during the COVID Pandemic in Florida Using a 3-D AccelerometerThis is a study evaluating the usefulness of a motion-detecting device to monitor pruritus in dogs in order to help adjusting the therapeutic plan. Although the device is unique and the purpose of the study is interesting, unfortunately, there are still some important questions not addressed in the design of the study (or not mentioned by the authors).

Overall, this article would be more appropriate for a journal presenting and explaining the purposes of motion-detecting devices. This study does not provide substantial scientific knowledge

Introduction:

One of the main concerns with this study is the fact that this electronic device has not been validated for the detection of pruritus. In other words, if the device detects a sleeping behavior or scratching, does the dog really sleeps or scratch?

If this method has been validated, it would be critical to mention this information (with references) in the introduction.

One objective I was expecting after reading the title and introduction was specifically the validation of the data recorded by this device. This would have been useful and relevant. If the authors compared the electronic data with owners’ observations, it must be indicated in the objectives.

Last paragraph, last sentence: I suggest adding that flare up are common in chronic pruritic skin diseases

Materials and Methods:

Again, the weakness of this study is to consider the data provided by the device as «true». There is no validation by owners of the behaviors recorded. Without this validation, we can’t give medical advice just based on those data.

Results:

Last sentence of the first paragraph (Owners were reluctant…) does not add value, I suggest deleting this sentence.

In this section, detailed results are lacking. The authors only show in Figure 1 an example of data.

In this section we should have the information regarding the population studied (age, breed, skin disease, etc.). This information could be summarized in a table.

There is no information regarding what the devices recorded. What is the average time of sleep? How much time on average per day dogs are grooming? Etc.

There is no information either on the medical issues. On average, how often do the devices detect pruritus? How many times has pruritus detection enabled the therapeutic plan to be adjusted? In how many dogs? Was there any relationship between the detection of pruritus and specific signs of skin disease detected by the medical team (e.g. otitis, excoriations, bacterial pyoderma…)?

This relevant information would give some content and value to this article.

Discussion:

In the first sentence, the authors mention that this device is valuable. But unfortunately, I can’t see, based on the results presented, how they can conclude or assume this is a valuable device.

It is mentioned «… daily data summaries for each dog were compiled into a single line chart for the full four-month monitoring period showing scratching, shaking, grooming and sleeping trends. These charts were created at the end of the study.» … « Each patient had a baseline pattern of scratching or shaking which could show an increased amount of activity that is characteristic for that dog but different from other dogs ». However, these results are not presented in the results section.

I suggest moving the 6th paragraph (Each daily ANIMO… previous week of data) in the M&M.

There is no mention of the limitations of this study (accuracy of the device, the ease with which owners can use the device and the application, …).

Conclusions:

Considering the design of this study, I think that the authors should be careful not to ‘overinterpret’ the results, especially as regards pruritus monitoring.

Last sentence: The authors have not really demonstrated that this electronic device is a valuable tool for the management of chronic pruritic skin disease. I would recommend rephrasing this sentence to be more aligned with the design if this study.

Author Response

Response to Reviewer 3

By Robert Lavan

For a manuscript entitled :

“Remote Monitoring of Canine Patients Treated for Pruritus during the COVID Pandemic in Florida Using a 3-D Accelerometer”.

Authors:

Introduction:

  1. One of the main concerns with this study is the fact that this electronic device has not been validated for the detection of pruritus. In other words, if the device detects a sleeping behavior or scratching, does the dog really sleeps or scratch?

If this method has been validated, it would be critical to mention this information (with references) in the introduction.

Response of the authors: The device is validated for specific behaviors that generate alerts and these behaviors are listed with ANIMO. There is not one behavior that is pathognomonic for pruritus, e.g. scratching and shaking may or may not be shown by an itchy dog. Licking may also be a sign of itch but ANIMO is not validated for licking. The current manuscript is the first proposed publication for this device and information on the validation process was added to the manuscript.

  1. One objective I was expecting after reading the title and introduction was specifically the validation of the data recorded by this device. This would have been useful and relevant. If the authors compared the electronic data with owners’ observations, it must be indicated in the objectives.

Response of the authors: See the previous answer regarding validation studies performed by Sure Petcare comparing video recordings and owner notes against dog behaviors demonstrated. The current study used the opposite methodology. Owners were contacted by the veterinary hospital when the device provided alerts regarding increased scratching, itching, grooming, etc. Owners were then asked about their observations and requested to indicate whether they thought that veterinary intervention was needed.

  1. Last paragraph, last sentence: I suggest adding that flare up are common in chronic pruritic skin diseases
    Response of the authors: The last sentence in the Introduction already carries this concept. We added this concept to the last paragraph, last sentence in the Conclusion.

Materials and Methods:

  1. Again, the weakness of this study is to consider the data provided by the device as «true». There is no validation by owners of the behaviors recorded. Without this validation, we can’t give medical advice just based on those data.

Response of the authors: The device was validated prior to becoming commercially available. The process used by Sure Petcare was the same used by all other 3-D accelerometer manufacturers and uses video recordings to develop algorithms for specific detectable behaviors. Note that the intention is not to use the data to give medical advice but to alert veterinarian and pet owner that a condition meriting veterinary examination was potentially present.

Results:

  1. Last sentence of the first paragraph (Owners were reluctant…) does not add value, I suggest deleting this sentence.

Response of the authors: We disagree. Getting pet owners to bring their animals back to the veterinary clinic is already a challenge. At the time this was compounded by the societal fear of COVID virus exposure risk, and a COVID vaccine was not yet available. This manuscript describes the value of a remote monitor to supplement veterinary care while the pet remains at home.

  1. In this section, detailed results are lacking. The authors only show in Figure 1 an example of data.

Response of the authors: Thank you for this observation. We included Tables 2 and 3 to demonstrate the average data obtained.

  1. In this section we should have the information regarding the population studied (age, breed, skin disease, etc.). This information could be summarized in a table.

Response of the authors: We added Table 2 and the second paragraph in Results to address that.

  1. There is no information regarding what the devices recorded. What is the average time of sleep? How much time on average per day dogs are grooming? Etc.

Response of the authors: Agree. Added Table 3 for just this reason.

  1. There is no information either on the medical issues. On average, how often do the devices detect pruritus? How many times has pruritus detection enabled the therapeutic plan to be adjusted? In how many dogs? Was there any relationship between the detection of pruritus and specific signs of skin disease detected by the medical team (e.g. otitis, excoriations, bacterial pyoderma…)? This relevant information would give some content and value to this article.

Response of the authors: This 3D-accellerometer is not diagnosing disease and pruritus detection is not yet measurable. The device detects movements that are validated as indicating specific behaviors identified through video studies and then characterized using algorithms to interpret data from the device. The clinic reviewed all alerts provided for all enrolled dogs then called owners whose dogs had alerts to ask for their observations and discuss therapy adjustment or recheck visit depending on these observations. Owners reporting no concerns were advised to watch and wait and call back if any indication of adverse clinical sign recurrene.

Discussion:

  1. In the first sentence, the authors mention that this device is valuable. But unfortunately, I can’t see, based on the results presented, how they can conclude or assume this is a valuable device.

Response of the authors: That statement came from Dr Mike Canfield and his staff as well as the pet owner survey conducted at the end of the study. For example, 90% of dog owners who completed the exit survey would recommend ANIMO to a friend.

  1. It is mentioned «… daily data summaries for each dog were compiled into a single line chart for the full four-month monitoring period showing scratching, shaking, grooming and sleeping trends. These charts were created at the end of the study.» … « Each patient had a baseline pattern of scratching or shaking which could show an increased amount of activity that is characteristic for that dog but different from other dogs ».However, these results are not presented in the results section.

Response of the authors: Thank you for this comment. We realized that we needed to show the different ways that line charts could appear and added this to the Results section.

  1. I suggest moving the 6thparagraph (Each daily ANIMO… previous week of data) in the M&M.

Response of the authors: Agree. Moved the paragraph to Methods and Materials.

  1. There is no mention of the limitations of this study (accuracy of the device, the ease with which owners can use the device and the application, …).

Response of the authors: Animo provided alerts indicating movement outside a specific range validated in previous work. This study recorded these alerts from the device worn on client owned dogs known to be at risk for pruritic dermatologic flare up. This methodology does not produce results that an be described using calculations of "accuracy” or “precision”. A dog may or may not show a particular individual behavior pattern in a given clinical situation (ie> scratching when a flea infestation starts). An exit survey was used to understand pet owner experience with the tool which includes the level of engagement, favorite features, recommendation to other dog owners and whether it was a good experience or not. 96% of owners who completed the study said it was a favorable experience and 90% would recommend it to a friend.

Conclusions:

  1. Considering the design of this study, I think that the authors should be careful not to ‘overinterpret’ the results, especially as regards pruritus monitoring.

Response of the authors: The authors do not propose Animo as a diagnostic tool. This study, used the ANIMO device as a tool for potential detection of behavior changes that could be related to pruritic flare-ups. The action taken was to increase communication between veterinarian and pet owner and potentially initiate an earlier return to the hospital for examination than might have been the case without this additional information. The paper presents examples showing the potential for earlier intervention with effective communication between veterinarian and pet owner around potential indications of movement changes in dogs with pruritic health conditions.

  1. Last sentence: The authors have not really demonstrated that this electronic device is a valuable tool for the management of chronic pruritic skin disease. I would recommend rephrasing this sentence to be more aligned with the design if this study.

Response of the authors: Again we disagree. ANIMO adds a layer of monitoring onto the veterinary therapeutic plan. Prior to the use of 3D-accellerometers, there was no monitoring except what the pet owner observed. Many dogs learn that they get punished when the owner sees them scratching and may scratch more intensively when not under immediate observation. The ANIMO is always there and backs up the owner’s observations.

Reviewer 4 Report

Comments and Suggestions for Authors

Author Response

Response to Reviewer 4

By Robert Lavan

For a manuscript entitled :

“Remote Monitoring of Canine Patients Treated for Pruritus during the COVID Pandemic in Florida Using a 3-D Accelerometer”.

Authors:

  1. The article proposes the use of a device that can be very useful to objectify dog's pruritus and maybe help to monitor treatment or even underlying causes. Discussion and conclusion seem to show that it is easy to use, has minimal impact on the animal, and is effective in its primary aim: monitoring pruritus manifestation. But the only results given are those of the exit survey. The accuracy of the device, how the charts were developed, how the vet medical staff actually used the daily results, etc. are not covered by the study material and methods nor the results.

Response of the authors: Thank you for your comments. Our proposal is that the device can be an “early warning” leading to better communication between owner and veterinarian and we do not propose use to monitor treatment or diagnose underlying causes. We rethought how we address the impact and included tables 2 and 3 to show both the population demographics and the average values across the four skin diagnosis categories. We also expanded our discussion of the line charts to show the different characteristics for dogs that generated small,  medium or large waves in the study. The paper presents how veterinary hospital staff used “alerts” to call owners regarding any changes observed in their dogs’ behaviors, that could indicate need for a recheck visit, a change in therapy or further observation.

  1. In the title and the discussion, the covid-19 period is mentioned as if an ideal moment to use such a device, but the results show that less device where implemented at that period of time.

Response of the authors: This question is unclear. ALL DOGS enrolled in this study wore the ANIMO device during their participation.

  1. My concerns are also related to the absence of ethical committee authorization, and the subsequent insufficient consideration of ethical issues. I agree that the device has minor to no effect of the dog, as most of them are used to wear collars. - The owner is completely free to stop the experiment at any time. Indeed the owner signs a consent that states, among other obligations: "If such withdrawal is without cause, you hereby agree to reimburse MAH for the actual costs of purchase of the Animo Device."

Response of the authors: IACUC approval is not required for non-laboratory (field) studies where the natural behavior of animals is being unobtrusively observed. This was a non-interventional cohort study of animals coming to veterinarians for care. The statement about study withdrawal “without cause” refers to pet owners who take the ANIMO without completing the study. All pet owners who completed the study had the option of keeping the ANIMO for their pet free of charge.

“Studies that involve unobtrusive observation of animals in their natural habitats do not require IACUC oversight. If the study has a potential to cause harm or materially alter the behavior of the animals, then IACUC oversight is required.”

https://research.uci.edu/animal-care-and-use/do-you-need-iacuc-review/#:~:text=Studies%20that%20involve%20unobtrusive%20observation,then%20IACUC%20oversight%20is%20required.

Do You Need IACUC Review? - UCI Office of Research

  1. Another issue related to this obligation is that the consent does not precise what is considered "without cause" and who decides what is considered a valid cause or not. - how owners' data will be used and how they can have access to their data is not described.

Response of the authors: A legally reviewed owner consent form was signed by all owners participating in this study. This document is not included in the report and it described all data use and data sharing that can and cannot occur and is aligned with usual field study practice and patient confidentiality.

  1. In the material and methods, I read no procedure to render data anonymous before they are handled by the veterinary clinic to bodies outside the owner to veterinary team bound, such as Merck/Intervet laboratory, FDA, etc.

Response of the authors: No individually identifiable data from this study are shared with any other party and owner confidentiality is protected. Veterinary medical data (outside of demographics) stays within the records of the veterinary clinics and the signed consent agreement includes confidentiality restrictions.

  1. - the use of the data by the company that developed the device is not mentioned nor described - regarding animal's data, they are treated as of commercial objects data.

Response of the authors: This question is unclear and the privacy of dog owners using the Animo device is protected.

  1. - After the study, both dogs and owners are left with the device, but apparently, beside the knowledge acquired during the study, no support/advice is made available for them. Even though without them the study would not have been possible, and the device could not have been developed not commercialize, once the study is finished, they are left alone, unless, I imagine, they continue to visit and pay consultations to the clinic.

Response of the authors: These assumptions are incorrect. Owners can call the veterinary hospital or an ANIMO customer service line with any request when helpis required. Dog owners are not required to pay an ongoing subscription and dog owners in this study received the ANIMO as theirs to keep at the end of the study with no cost to them. If an owner did not complete the study period, then they were asked to return the ANIMO per the agreement.

  1. - the extend to which such data will be use for the sake of animal health itself, and not only for commercial purposes. For instance, following these dogs yearly for some weeks could give insight into the natural evolution of the disease. Such study probably has little interest from the device commercial development point of view, but has a lot on knowledge about naturally occurring AD.

Response of the authors: Thank you for this suggestion and we are also considering ideas for future projets.

  1. Other ideas of study for the sake of animals and for helping owners, often truly distressed by the daily care, could be done related to this device. I understand that such concerns may seem idealistic, and that there are cultural and legislative differences. Nevertheless, they are more and more considered necessary by the general population to make studies on animals acceptable.

Response of the Authors: See response above.

  1. Also, scientific publication and activity also have effects on improving the way both humans and animals are treated.

Response of the Authors: See response above.

  1. - researchers have access to data important regarding the dogs that are denied to the owners: "SPC is also evaluating a new experimental variable (time spent grooming) which was not included in the pet owners report but was available to the researchers" Line 96-97. Are there other types of data available to researchers/engineers and not to owners? Apparently, no GPS tracking, any other? Are both the veterinarians and the pharmaceutical company sure that none of the trackers were performing GPS tracking or other undisclosed features?

Response of the Authors: None of the Animo devices in this study had GPS tracking or other undisclosed capabilities.

  1. Regarding the monitoring, more precise details should be provided on how each parameter is monitored, what choices (both technical and medical) were made, etc. In the text or also in supplemental material.

Response of the Authors: Table 1 and 3 lists the variables captured along with the units of measure. Each parameter has a previously validated proprietary algorithm. Validation assures that movements recorded match to actions such as resting, sleeping, scratching, or shaking. 

  1. Line 89 -91: "The ANIMO device is a small (1.45 inch diameter × 0.45 inch depth; 0.78 oz) round" "to 3.3 feet": Please use (or add) international measurement (meters/cm, grams), and in the rest of the article (collar width, etc).

Response of the Authors: We added the international measurements.

  1. Table 1 : - grooming is apparently a variable provided by ANIMO. So the content of the table is not coherent with its name.

Response from the Authors: The grooming variable was added to the table with a superscript indicating at the time of the study, it was being considered for inclusion.

  1. - Is "Wellbeing" totally equal to "scratching and shaking per day" ? Seems quite reductive view of dog's wellbeing. - scratching and shaking are assessed only by the total of minutes/day. Are the 'bouts' also an available data?

Response from the Authors: “Wellbeing “describes the “scratching” plus “shaking” variables. “Grooming” may be included in this group in future. Scratch and shake alerts are not included within “wellbeing” and are an indication of a more intense movement response. The average dog generated about 1 alert per week (or 16 alerts over 17 weeks).

  1. Line 141: what is considered to be the night? fixed hours (eg. 10PM to 6AM)? Sunset to sunrise?

Response from the Authors: Owners were asked to define this period for their household during initial set up.

  1. More generally, more precise details should be provided on how each parameter is monitored, what choices (both technical and medical) were made, etc. In the text or also in supplemental material.

Response from the Authors:  This is a duplicate of question 12 above.

  1. 155-162 : " Each morning, the app presented owners of dogs with the ANIMO device a summary 155 of the data generated. The study team also provided a summary of the previous weeks’ 156 monitoring data to the veterinary clinic staff. This summary included: a list of scratch and 157 shake alerts, notification when an owner was delayed in syncing their dog’s data, updates 158 on devices needing a battery change, and reports of dogs not wearing the ANIMO device. 159 Clinic staff used this weekly report to review the status of enrolled dogs and determine 160 whether a call to the owner was needed for an update or action. When a scratch or shake 161 alerts was recorded, owners were asked if they were seeing increased scratching or shak- 162 ing and then had an opportunity to set up a recheck visit and/or adjust therapy. " Most of what is exposed in this § belong to material and methods.

Response from the Authors: We agree.  This was moved to Materials and Methods.

  1. Line 164 - 167 " An example is included to show how information from the app flagged an increase 164 in itching. A male neutered 9 your old 6 kg pug cross dogs was previously diagnosed with 165 atopy that was being treated at home. ANIMO provided warning of a flare up in his pru- 166 ritic condition before this was observed by the owner. Communication between the vet- 167 erinarian and owner was initiated leading to modification of the pruritis management 168 protocol and improvement in clinical signs." It is interesting to present an exemple, so the reader understand better, but this is not scientifically relevant on its own.

Response from the Authors: Clinical veterinarians have only recently gained access to remote monitoring devices that can supplement pet owner observations. This paper presents anecdotal observations of how such devices can be used based on experience with Animo and discussed potential utility and benefits for the pet owner.

  1. Line 170 - 176 " Line charts were developed by the study team to visually represent 4 months of in- 170 formation. This information could be compiled from the ANIMO App by examining data 171 fields on consecutive days. Specific calendar dates were aligned with study days (day 1 to 172 day 120) to allow association between events in the medical notes (communication with 173 the pet owner, for example) and activities in the 4-month line chart. The wave amplitude 174 for scratching, shaking and grooming behavior and the number of shake or scratch alerts 175 was unique to each dog but remained consistent for individuals during the 4-month mon- 176 itoring window. " material and methods

Response from the Authors: Agree. Moved to Materials and Methods.

  1. Results mainly consist on the exit survey. This is not coherent with the title ("Remote Monitoring of Canine Patients Treated for Pruritus during the COVID Pandemic in Florida Using a 3-D Accelerometer "). The title : 1/ focuses on the device. I would expect how the device helped the vet and the owner to treat the animal and monitor it 2/ mention the covid pandemic, but in the article the only result related to the pandemic is that owners were reluctant to come, so the study was slow. The content of the exit survey should be presented in the complemental material.

Response from the Authors: We restructured the Materials and Methods following your suggestions and created additional material for the results section, which focused on the device and how it was used in this study.

  1. Line 190 " Accelerometer movement monitoring by a collar-attached device provided valuable 190 medical management data for dogs presented with pruritus to a veterinary dermatologi- 191 cal referral practice." Is this a result, then back it on the observation from the study. If it is a general comment, add scientific literature references.

Response from the Authors: Several co-authors of this paper (M Canfield, T Canfield,  J Thomas, B Sampeck and T Springer) were clinical staff members who worked with these patients and conducted the study. The authors wrote that phrase (and the rest of the manuscript) because of direct experience with the ANIMO. The phrase you call out is a legitimate observation from the clinic staff who also serve as co-authors for this manuscript. No literature reference is needed.

  1. Line 193 - 194: only one exemple is conforting that result. " In 193 particular, the potential to provide an early alert of pruritus “flare up” offered notable 194 value." Line 196 " This study was conducted against the background of a global pandemic that began 196 in 2020 and continues into 2022, at the time of writing. This timing was unexpected and 197 coincidental; however, provided a unique opportunity to evaluate a remote movement 198 monitoring device to follow therapy for dogs being treated for pruritic skin disease." This could have been interesting, but these comments are not supported by any result presented. Moreover, the pandemic induced less device implemented.

Response from the Authors: We don’t know what you mean by “the pandemic induced less device implemented”. All dogs accepted into the study had an ANIMO on their collar and pet owners were very willing to participate in the study.

  1. Line 210 : " reduced nightly sleep quality, daily energy expenditure " : any dog GPS that cost a small fraction of the price gives these data.

Response from the Authors: This question is unclear and GPS linked devices were not used in this study.

  1. Line 214: " On one case, the clinic staff 214 called the pet owner to inquire about the dog’s episodes of interrupted sleep. The owner 215 reported that the dog was getting up every time she experienced insomnia and the app 216 was able to detect this. " This result has not been exposed in the result part of the article. Is it substantiated or is it only a single story?

Response from the Authors: This anecdote was related by an enrolled dog owner when the clinic called to ask about device data recording the dog having interrupted sleep.

  1. Most of the discussion is general on what can be expected from such a device, and thus belongs to the introduction. This would allow the material and methods to be adapted so that the results in the article is more related to what can be expected the article's title.

Response from the Authors: Additional explanation of the variables along with average values for parameters are now included.

  1. The trade-name of the device is mentioned from 2 to over 16 times/page. This, plus the quasi absence of results beside owner satisfaction, question the aim of this article as being a communication on the device or as being a scientific article.

Response from the Authors: The ANIMO device does not have a nickname which could be substituted, nor does it belong to a class of devices currently used in veterinary practice. The use of its proper name should be logical if some aspect of that specific device is being described. Legitimate results are shown which demonstrate the kinds of data that could be captured. The aim of the manuscript is to inform pet owners, veterinarians and their staff on the capabilities of a very new technology.

  1. Conclusion " Accelerometer use provides an additional layer of monitoring for pruritic skin dis- 279 ease especially when close monitoring is difficult or neglected. The use of a remote moni- 280 toring device allows veterinary hospital staff to see how dog owners are progressing with 281 managing their dog’s pruritus at home. The ability of the dog owner, clinic staff and/or 282 veterinarian to observe the amount of daily calories burned, sleeping, scratching, shaking 283 (and grooming when available) provides a more complete picture on the dog’s response 284 to therapy. Recording exercise also drives many dog owners to increase their own exercise 285 when they increase the exercise of their pets to meet daily goals. Use of movement moni- 286 toring devices are a potentially valuable new aid for the management of chronic pruritic 287 skin disease of dogs. " In order to substantiate such a conclusion, the study should have explored whether the device accurately monitor what is says it is monitoring. If a mouvement to catch a fly is not misinterpreted as a scratch, etc.

Response from the Authors: This study recorded alerts received based on previously validated movement detection. This study did not undertake further validation. A very short action – such as catching a fly – would be insufficient to generate an alert.